# Salt Priming as a Smart Approach to Mitigate Salt Stress in Faba Bean (*Vicia faba* L.)

**DOI:** 10.3390/plants11121610

**Published:** 2022-06-20

**Authors:** Amira K. Nasrallah, Mohamed A. M. Atia, Reem M. Abd El-Maksoud, Maimona A. Kord, Ahmed S. Fouad

**Affiliations:** 1Botany and Microbiology Department, Faculty of Science, Cairo University, Giza 12613, Egypt; anasrallah@sci.cu.edu.eg (A.K.N.); kord@sci.cu.edu.eg (M.A.K.); 2Genome Mapping Department, Agricultural Genetic Engineering Research Institute (AGERI), Agricultural Research Center (ARC), Giza 12619, Egypt; 3Nucleic Acid & Protein Chemistry Department, Agricultural Genetic Engineering Research Institute (AGERI), Agricultural Research Center (ARC), Giza 12619, Egypt; reem.mohsen@ageri.sci.eg

**Keywords:** salt priming, salt stress, gene expression, qPCR, *Vicia faba*, antioxidant enzymes

## Abstract

The present investigation aims to highlight the role of salt priming in mitigating salt stress on faba bean. In the absence of priming, the results reflected an increase in H2O2 generation and lipid peroxidation in plants subjected to 200 mM salt shock for one week, accompanied by a decline in growth, photosynthetic pigments, and yield. As a defense, the shocked plants showed enhancements in ascorbate peroxidase (APX), catalase (CAT), glutathione reductase (GR), peroxidase (POX), and superoxide dismutase (SOD) activities. Additionally, the salt shock plants revealed a significant increase in phenolics and proline content, as well as an increase in the expression levels of glutathione (GSH) metabolism-related genes (the L-ascorbate peroxidase (L-APX) gene, the spermidine synthase (SPS) gene, the leucyl aminopeptidase (LAP) gene, the aminopeptidase N (AP-N) gene, and the ribonucleo-side-diphosphate reductase subunit M1 (RDS-M) gene). On the other hand, priming with increasing concentrations of NaCl (50–150 mM) exhibited little significant reduction in some growth- and yield-related traits. However, it maintained a permanent alert of plant defense that enhanced the expression of GSH-related genes, proline accumulation, and antioxidant enzymes, establishing a solid defensive front line ameliorating osmotic and oxidative consequences of salt shock and its injurious effect on growth and yield.

## 1. Introduction

Faba bean is one of the most important winter legume crops cultivated worldwide. It is a versatile crop used for human food, animal feed, pharmaceuticals, and industrial purposes [1]. Its seeds are a high-quality source of protein (18–35% of dry matter) [2], energy (320 Kcal/100g) [3], vitamins (such as thiamin, riboflavin, pyridoxine, folic acid, and vitamins E and K) in amounts less than 1 mg, and antioxidants (including phenols, flavonoids, and anthocyanin) which are found in variable amounts in the plant parts [3,4]. In addition, the legumes provide a valuable source of L-DOPA, a precursor to a Parkinson’s disease drug [5]. The roots also enhance soil fertility through symbiotic union with *Rhizobium* bacteria, which enriches soil nitrogen [6].

Salinity stress is a major abiotic stress that dangerously influences crop plant growth and development, from seed germination to yield [7]. It is responsible for a 30% loss in global food production [8]. Globally, it is estimated that about one billion ha are impaired by soil salinization and that soil salinization is expanding, at an annual rate of 2 million ha [9]. About 20% of arable lands and more than 7% of total land areas are affected by salinity, especially in arid and semi-arid regions, such as Egypt [10]. Moreover, it is anticipated that most of the world’s arable lands will be salinized by the middle of this century [11].

Salinity influences all the physiological processes in the plant body, including respiration, photosynthesis, lipid metabolism, and protein synthesis [12]. Initially, salt stress reduces the rate of leaf expansion, with a consequent significant decrease in shoot growth [13]. The injurious effects of salt stress on plant growth primarily originate from the accompanying ionic toxicity and osmotic stress [14]. Sodium and chloride toxicity interrupt plant nutrition and induce physiological drought by decreasing the osmotic potential of the soil solution [15]. Furthermore, elevated concentrations of sodium ions (Na^+^) provoke the enormous generation of reactive oxygen species (ROS), including hydrogen peroxide (H_2_O_2_), superoxide ions (O_2_^−^), hydroxyl radical (OH^−^), and singlet oxygen (^1^O_2_) in plant cells, which are accompanied by lipid peroxidation, membrane damage, altered levels of growth regulators, nutrient imbalance, enzymatic and metabolic disorders damaging photosynthetic activities, and, finally, plant death [16].

As a defense, plants accrue some compatible solutes, such as proline, sugars, amino acids, and proteins [17], which implement water absorption [18]. These solutes participate in osmotic adjustment in plant cells, which preserves cellular homeostasis [19]. In addition, oxidative stress activates and orchestrates enzymatic and non-enzymatic antioxidants to tramp the accrued ROS [20]. Superoxide dismutase (SOD) is the anterior defense line against oxidative damage [21]. It dismutases (O_2_^−^) into H_2_O_2_ and molecular oxygen [22]. Catalase (CAT), in turn, converts H_2_O_2_ into H_2_O and O_2_ [23]. Additionally, peroxidase (POX) oxidizes various phenolic substrates with the synchronized conversion of H_2_O_2_ into H_2_O and O_2_ [24]. Ascorbate peroxidase (APX) also detoxifies H_2_O_2_, using ascorbic acid as a reducing agent with the generation of H_2_O and MDHA (monodehydroascorbate) [25]. At the same time, metabolism is directed toward the biosynthesis of reduced compounds, such as phenolics [26].

Glutathione (GSH) is a ubiquitous tripeptide (γ-glutamyl-cysteinyl-glycine; γ-Glu-Cys-Gly); the thiol group of Cys residue is the reactive group responsible for the biochemical and biological functions of GSH [27]. It participates in ROS detoxification through the ASA-GSH cycle [28], during which GSH is oxidized into glutathione disulfide (GSSG), which regenerates GSH with the aid of glutathione reductase (GR) [29]. However, there is growing evidence of GSSG degradation into Glu and Cys–Gly by γ-glutamyl-transferases (GGTs; E.C. 2.3.2.2) and further degradation of Cys–Gly into Cys and Gly by members of the M17 family of metallopeptidases, including leucyl aminopeptidase (EC 3.4.11.1) and aminopeptidase N (EC 3.4.11.2); the free amino acids are used in the re-synthesis of GSH [30,31].

In addition its role in ROS detoxification through the ASA-GSH cycle, two GSH molecules covalently attach to the spermidine (Spd) molecule, yielding trypanothione (T(SH)_2_) in a reaction catalyzed with trypanothione synthetase [31]. T(SH)_2_ plays a crucial role in many cellular functions, including the detoxification of H_2_O_2_ and the indirect synthesis of deoxyribonucleotide through the activity of ribonucleotide reductases [31,32].

Despite these defensive mechanisms, the plant’s productivity is reduced dramatically under salt stress. Therefore, several modern approaches in the last decade focused on promoting the ability of the plant to mitigate salinity damage, including the external application of salts; this process is referred to as the plant’s adaptation to salt stress [11]. “Priming” is exposing a plant to a mild dose of stress to activate a stress recovery memory [33]. Moreover, priming induces stress tolerance mechanisms by activating some physicochemical responses, such as antioxidant scavenging systems [34], which make the primed plants capable of quicker and more effective stress-protective responses [35,36]. Seeds or seedlings can be primed; seed priming is a controlled soaking technique for treating seeds with natural and synthetic substances before germination to improve imbibition, rapid germination, and final yield [37]. Similarly, plant priming is a method of hardening plants by subjecting them to initial stressors that enable the primed plants to initiate defensive processes faster than unprimed plants [38,39].

Numerous studies have shown that priming with salt improves plant resistance to salt stress on different crops by osmolyte accumulation and decreasing ionic toxicity [33,40]. In addition, by reprogramming the expression of relevant genes, priming establishes the building up of certain osmolytes and increases the activity of numerous antioxidant enzymes, protecting the cell against salinity-induced damage [41]. However, there are no publications concerning the impact of salt priming on the expression of glutathione-metabolism-related genes.

Therefore, the present investigation aims to study the potential role of salt priming in alleviating salinity stress in faba bean, focusing on the expression of some glutathione metabolism-related genes, including the *L-APX*, *SPS*, *LAP*, *AP-N*, and *RDS-M* genes. In addition, the role of salt priming in the activation of enzymatic and non-enzymatic antioxidants and the accumulation of compatible solutes are highlighted.

## 2. Results

### 2.1. Plant Growth and Yield

Compared with the NC plants, the PC plants showed a significant decrease in all the considered growth parameters, ranging from 19.4% to 47%, recorded for plant height and shoot dry weight, respectively (Figure 1 and Appendix A). Salt priming had no significant effect on the shoot fresh weight and the leaf area, compared with the same control. On the other hand, P plants exhibited 18.2% and 7.7% decreases in shoot dry weight and plant height, respectively. At the same time, the root fresh and dry weights increased by 15% and 32.1%, respectively. After 7 days of 200 mM salt shock, the root dry weight of the Shk group remained insignificantly affected, compared with the NC plants. However, the salt shock was accompanied by a significant decrease in the remaining parameters, ranging from 14.4% to 30.6%, measured in root fresh weight and shoot dry weight, respectively. Salt priming did not modify the influence of salt shock on the shoot fresh weight. Conversely, the shoot dry weight (13.8%), root fresh weight (25.1%), root dry weight (10.3%), leaf area (23.1%), and plant height (12.6%) reflected significant increases in P/Shk plants, compared with Shk plants. The alleviatory effect of priming was most apparent, monitoring the roots where priming nullified the inhibitory role of salt shock.

Regarding yield-related traits, results reflected a 39% to 51% decrease for the weight of 1000 seeds and seed weight per plant, respectively, in the PC plants, compared with the NC plants (Figure 1 and Appendix A). Salt priming had no significant effect on yield-related traits, compared with the NC plants, except for pod length, which showed an 8% decrease in the P plants. Yield-related traits reflected a significant decrease for pod length (19.4%), pod weight per plant (15.9%), the weight of 1000 seeds (22.8%), and seed weight per plant (24.39%) in the Shk plants, with respect to the NC plants. Conversely, pod length (12.6%), pod weight per plant (10.8%), the weight of 1000 seeds (18.3%), and seed weight per plant (16.4%) reflected a significant increase in the P/Shk plants, in comparison with the Shk plants. Priming completely nullified the impact of salt shock in regard to pod weight per plant.

### 2.2. Chlorophyll and Carotenoids

The total chlorophyll and carotenoids in the PC plants decreased, reaching 81.4% and 74.3%, respectively, of the NC plants at the end of the priming period (Figure 2 and Appendix A). Replacing the PC plants with P plants resulted in a 16.4% decrease in carotenoids, while chlorophyll remained insignificantly affected.

At the end of the shock period, the chlorophyll content showed a 14.6% decrease in the PC plants, compared with the other control, while the carotenoids remained unaffected. The Shk plants reflected a significant decrease in chlorophyll (27.9%) and carotenoids (26.9%), respectively, compared with the NC plants. Compared with the same control, priming alone had no significant effect on either pigment and nullified the inhibitory effect of salt shock.

Compared with the NC plants, the carotenoids/chlorophyll ratio showed a 7.6% and 16.7% decrease in the PC plants and the primed plants, respectively, at the end of the priming period. Conversely, the ratio reflected a 10% increase in both treatments at the end of the shock period, compared with the corresponding NC plants. Compared with the same control, the salt shock had no significant impact on the carotenoids/chlorophyll ratio in unprimed plants, while it was accompanied by an approximate increase of 5% in primed plants.

### 2.3. Lipid Peroxidation

Compared with the NC plants, lipid peroxidation, as indicated by increased MDA content, increased by 44.4% in the PC group at the end of the priming period (Figure 3a and Appendix A). At the same time, priming was associated with an 18.5% increase. At the end of the salt shock period, lipid peroxidation reached 1.58- and 1.19-fold in PC plants and P plants, respectively, compared with the NC plants. On the other hand, salt shock resulted in 53% and 44% increases in lipid peroxidation in unprimed and primed plants, respectively.

### 2.4. Hydrogen Peroxide Content

H_2_O_2_ accumulation increased by 53% and 36% at the end of the salt priming period in PC plants and P plants, respectively, compared with the records for NC plants (Figure 3b and Appendix A). The PC and P groups were associated with 74.7% and 29.5% increases, respectively, compared with the corresponding NC plants, at the end of salt shock period. Records from the Shk plants reflected that salt shock resulted in a 69.4% increase in H_2_O_2_ content; however, priming decelerated the H_2_O_2_ accumulation related to salt shock.

### 2.5. Membrane Stability Index (MSI)

At the end of the salt shock period, the PC plants showed a 6.4% decrease in MSI, compared with the NC plants; conversely, salt priming was accompanied by a 2.4% increase (Figure 4 and Appendix A). The salt shock was associated with a 9.7% decrease in MSI for the NC plants. Priming completely mitigated the injurious effect of salt shock on membrane stability, which remained comparable with the NC value.

### 2.6. Phenolic Content

Compared with the NC plant results, the total and free phenolics showed a 31% and 6.6% increase, respectively, in the PC plants at the end of the priming period (Figure 5 and Appendix A). Replacing PC plants with P plants reflected 53.4% and 40% increases for phenolics attributes.

At the end of the salt shock period, the total and free phenolics measured in the PC plants reached 106% and 119% of the corresponding NC values. At the same time, the P plants exhibited 33.3% and 63.6% increases for total and free phenolics, respectively. On the other hand, Shk plants accumulated 42% and 72% higher amounts of total and free phenolics, respectively. Priming decreased the accumulation of total phenolics in the P/Shk plants, reaching 83% of the Shk group. The same comparison reflected an 18% increase in free phenolics accumulation.

### 2.7. Enzymatic Antioxidant Activities

At the end of salt priming period, the monitoring the enzymatic antioxidant activities—POX (81%), CAT (41%), GR (48%), APX (41%), and SOD (51%)—reflected increases, in the PC plants, compared with the NC plants (Figure 6 and Appendix A). The same enzymes—POX (87%), CAT (122%), GR (82%), APX (122%), and SOD (51%)—showed significant increases in primed plants.

Compared with the corresponding NC plants, the promoting effect related to exposure to salt stress was also demonstrated in antioxidant enzymes during the salt shock period. In the PC group, POX (91%), CAT (52%), GR (39%), APX (86%), and SOD (64%) activities reflected increases in the first 7 h of the salt shock period, which intensified at the end of the shock period. In the first 7 h of salt shock, POX (2.34-fold), CAT (2.27-fold), GR (1.65-fold), APX (2.22-fold), and SOD (1.68-fold) activities increased in P plants, compared with corresponding NC plants. The enhancements were decelerated for POX, CAT, and APX at the end of the shock period, while it was magnified for the remaining enzymes. Finally, exposure to salt shock for 7 h was accompanied by a significant increase in POX (181%), CAT (150%), GR (76%), APX (108%), and SOD (53%) activities in the Shk group. The enhancements were intensified upon extended salt exposure to 7 days for all enzymes except CAT, which retained the same promoting level.

Comparing the Shk group with the P/Shk group indicated that priming had no influence on the POX, GR, and SOD responses to 7 h of salt shock, while it amplified the response of CAT and APX. At the end of the salt shock period, priming modified the response of all enzymes to salt shock, except SOD, which remained insignificantly different in shocked plants independent of priming. While it intensified the response of GR and APX, priming decelerated the response of POX and CAT to extended salt shock.

### 2.8. Total Soluble Sugars

Comparison with the control irrigated with tap water indicated 42% and 59% increases in TSS regarding PC plants and P plants, respectively, at the end of the priming period (Figure 7a and Appendix A). However, at the end of the salt shock period, TSS reflected a 61% decrease in the PC plants, compared with the corresponding NC plants. On the other hand, salt priming had no significant effect on the TSS content in the P group. In addition, the salt shock only had no significant effect on TSS accumulation with no significant interaction with priming.

### 2.9. Free Proline Content

At the end of the priming period, the proline content increased by 75% and 130% in the PC plants and P plants, respectively, compared to the NC plants (Figure 7b and Appendix A). At the end of salt shock period, a comparison with the corresponding NC plants reflected 65% and 120% increases recorded in PC plants and P plants, respectively. The salt shock was accompanied by a 158% increase in proline content, which was significantly intensified by priming.

### 2.10. Expression of Glutathione Metabolism Genes

At the end of the salt priming period, the expression of the selected GSH-metabolism-related genes reached *L-APX* (1.45-fold), *LAP* (1.59-fold), *SpS* (1.45-fold), *AP-N* (1.8-fold), and *RDR-M* (1.3-fold) in the PC plants, compared with the expression of the corresponding genes in the NC group (Figure 8 and Appendix A). Simultaneously, the expression of these genes reached *L-APX* (1.7-fold), *LAP* (4.32-fold), *SpS* (3.07-fold), *AP-N* (4.66-fold), and *RDR-M* (2.48-fold) in the primed plants.

Seven h after starting the NaCl shock period, the expression levels in the PC plants reached 2.31-fold (*L-APX)*, 1.34-fold (*LAP)*, 1.69-fold (*SpS)*, 1.98-fold (*AP-N)*, and 1.73-fold (*RDR-M*) of the expression quantified in the corresponding NC plants. The stimulation was faded for the *L-APX, LAP* and *SpS* genes, while it was enhanced for the AP-N and *RDR-M* genes at the end of the shock period.

Results recorded for the P plants showed that priming provoked the expression of the addressed genes, reaching 1.59-fold (*L-APX*), 1.42-fold (*LAP*), 1.53-fold (*SpS*), 2.28-fold (*AP-N*), and 1.87-fold (*RDR-M*) for the corresponding gene expressions in the NC plants, 7 h after starting the NaCl shock period. In addition, the stimulatory effect of priming was intensified for all genes, except the *LAP* gene, whose expression reached 1.3-fold at the end of the NaCl shock period.

Salt shock for 7 h was associated with 1.15-fold (*L-APX*), 0.18-fold (*LAP*), 0.2-fold (*SpS*), 0.5-fold (*AP-N*), and 0.75-fold (*RDR-M*) increases in transcript abundance in the Shk group, compared with the corresponding NC group. Extending the shock to 7 days decelerated the expression of *L-APX* and *LAP*, while amplifying the impact of the salt shock on the remaining genes.

Priming downregulated the expression of GSH-metabolism-related genes recorded in salt shocked plants, with two synergetic exceptions, as shown, considering the Shk and P/Shk groups. The first exception was recorded for the *LAP* gene in the first 7 h of salt shock, while the second exception was observed for the *RDR-M* gene at the end of the salt shock.

## 3. Discussion

Salt stress always disturbs ion exchange, leading to ionic imbalance due to the synchronized Na^+^ influx and K^+^ efflux; the accumulated Na^+^ enhances ROS generation [16]. The results of the current study showed the oxidative stress demonstrated by the development of H_2_O_2_ as a consequence of exposure to salt treatment, either in the form of continuous irrigation with 150 mM of NaCl solution or salt shock with 200 mM of NaCl for one week. The accretion of H_2_O_2_ and other ROS clarifies the lipid peroxidation observed in this study, as shown by the increase in MDA in plants subjected to salinity stress. Boosted polyunsaturated fatty acid peroxidation diminishes membrane fluidity and enhances leakiness, leading to severe membrane deterioration [42] expressed directly as an increase in MSI [43]. The previous membrane-related issues are responsible for growth failure and declined pigment content recorded in the present investigation following salinity stress.

Nasrallah et al. [44] recorded the building up of H_2_O_2_ and MDA, accompanied by growth retardation and a decrease in pigments of faba bean foliage leaves, as a consequence of salinity stress. The same observations were recorded by Ahmad et al. [18] in pea exposed to salt stress. In addition, Frukh et al. [45] observed that the accumulation of H_2_O_2_ generated more MDA in rice plants under salinity stress. The relationship of an increase in H_2_O_2_ content and MDA accumulation was documented by several research groups [46,47,48].

The decline in total chlorophyll and carotenoid contents observed in plants subjected to salt stress is a familiar occurrence observed in several plants [49]. The inhibition of chlorophyll biosynthesis via provoking chlorophyllase enzymes and membrane deterioration are two possible explanations for this decrease [50]. The alteration in green photosynthetic pigment may be explained by oxidative stress, denaturation of the Rubisco enzyme [51], and modifications of the chlorophyll ultrastructure [52]. In addition to acting as an accessory pigment, carotenoids exhibit antioxidant properties; the enhanced carotenoids/chlorophyll ratio is a mechanism for avoiding photo-oxidation [53]. Under salt stress, an elevated carotenoids/chlorophyll ratio was recorded in sugarcane [54], *Leptochloa fusca* [53], maize [55], and faba bean [44].

On the other hand, the results of the present investigation reflected better growth and yield in P/Shk plants, compared with the results recorded in Shk plants. The alleviatory role of priming was also indicated in the growth, the photosynthetic pigments, and/or the yield of salt-primed shocked plants of sweet sorghum [56], wheat [11,57], and *Mesembryanthemum crystallinum* [58]. The better growth and yield of primed plants under salt shock can be attributed primarily to the ability of priming to culminate the oxidative stress symptoms, as indicated by the decrease in H_2_O_2_ and MDA, as well as by the increase in MSI. The role of priming in ameliorating salinity-accompanied oxidative stress was also documented in sweet sorghum [56] and wheat [11,57]. The aforementioned behavior of the primed plants can be ascribed to the observed role of salt priming in strengthening plant defense against salt shock, as detailed in the following sections.

As a potent oxidant, H_2_O_2_ induces a variety of injurious reactions in plants when it is converted into hydroxyl anion. Plants have an arsenal of enzymatic and non-enzymatic antioxidant mechanisms orchestrated in the identification, detoxification, eradication, and neutralization of the generated ROS to maintain cellular redox equilibrium [59]. SOD transforms superoxide radicals into oxygen and H_2_O_2_ [22]; the latter is decomposed into water and oxygen molecules through CAT activities [23]. POX contributes in the elimination of the overproduced H_2_O_2_ during the oxidation of various phenolic substrates [24]. The ascorbate–glutathione cycle is a crucial hydrogen peroxide scavenging mechanism in plant cells, in which APX plays a fundamental role in expediting the transformation of H_2_O_2_ to H_2_O and MDHA, utilizing ascorbates as electron donors [25]. Ascorbates are recovered through the ascorbate–glutathione cycle, governed by GR [60].

The present study reflected a boost in POX, CAT, GR, APX, and SOD activities in response to salt stress. In accordance with these results, Nasrallah et al. [44] documented the positive correlation between salt stress and the activities of POX, CAT, and SOD. The enhancements in these enzymes’ activities following exposure to 120 and 150 mM of NaCl were also assayed in pearl millet [61]. In addition, Ahmad et al. [62] recorded the elevation in CAT and the activities of SOD in tomato seedlings under salt stress. Moreover, Alnusairi et al. [43] documented a significant elevation in GR, APX, CAT, and SOD activities in salt-stressed wheat seedlings. The same results were recorded in *Phaseolus vulgaris* [63].

Compared with NC records, the current results reflected the increase in H_2_O_2_ in P plants at the end of the priming period, which endured following salt withdrawal. As a defense, our results mirrored a corresponding increase in the addressed enzymatic antioxidant activities. The correlation between the increase in H_2_O_2_ content and the enhanced activities of antioxidant enzymes was also documented by several research groups [44,47,64]. The enhancement of enzymatic antioxidant activities following priming was also recorded for SOD in salt-primed cucumber plants [65] and for SOD, as well as CAT, in *Zea mays* [66].

SOD activity, which is responsible for the dismutation of (O_2_^−^) into H_2_O_2_ and molecular oxygen [22], appeared to be comparable in both P/Shk plants and those of the Shk group. However, the detoxification of H_2_O_2_ in primed-shocked plants reflected bias toward APX assisted with GR, while the unprimed shocked plants favored CAT and POX.

Improvement in the antioxidant enzymes is a known explanation for the alleviatory role of salt priming against salinity-related oxidative stress [67]. However, the targeted enzymes seem to be genotype dependent and complicated with primed plant material. Working on sweet sorghum, Yan et al. [56] documented enhanced activity for SOD and APX in primed plants exposed to salt stress, compared with unprimed stressed conditions, while CAT appeared to be insignificantly affected with pretreatment. Working with the same species, Guo et al. [68] found, surprisingly, that seed priming with NaCl was accompanied by higher levels of CAT and POX under salt stress, compared with the exposure to salt stress without priming. The same results were recorded for melon plants derived from salt-primed seeds [69]. In a similar context, seed priming with H_2_O_2_ enhanced APX and CAT activities without a significant effect on SOD under salt stress, compared with unprimed stressed conditions [70].

The non-enzymatic antioxidants, exemplified in our results by phenolic compounds, accumulated in salt-stressed plants to reduce lipid peroxidation by trapping the lipid alkoxyl radical based on their structure [71]. Furthermore, phenolics have a proclivity to attach to the polar head of phospholipids, allowing them to accumulate at cell membranes and contribute to membrane integrity [72]. They prevent free radicals from diffusing into the membrane’s hydrophobic matrix and/or restricting oxidative stress-induced reactions in the bilayer’s hydrophobic region [73]. The increase in phenolics in response to salt stress was also recorded in faba bean [44], wheat [74], and rapeseed [75].

Our results reflected the role of salt priming in encouraging phenolics accumulation in P plants, which was also recorded in rapeseed [76] and *Calotropis procera* [77]. During salt shock, P/ShK plants synthesized lower amount of phenolics, with a simultaneous enhancement in hydrolysis of the conjugated phenolics, as indicated by the decrease in total phenolics and the increase in the free fraction, compared with the Shk plants. Thus, the enhanced salinity tolerance of primed plants may be ascribed to the role of salt priming in the reallocation of phenolics toward the free form, which is characterized by better antioxidant potential [76,78].

The monitoring of gene expression during salt stress reflected the upregulation of the *APX* gene, which confirmed the role of APX in mitigating salt stress. In agreement with our results, ElSayed et al. [63] documented a significant upregulation of the *APX* gene in *Phaseolus vulgaris* seedlings exposed to 200 mM of NaCl for 3 h and 14 days.

The increased *APX* gene expression, supporting the aforementioned increase in APX activity, accelerates the ASA-GSH cycle in regenerating ascorbates at the expense of GSH oxidation. The role of GSH in ROS detoxification through the ASA-GSH cycle was well documented by Foyer and Noctor [28], identifying the acceleration of mechanisms involved in the regeneration of GSH from its oxidized form as an important tool for salt tolerance.

In addition to the direct reduction of GSSG through the GR activity in the ASA-GSH cycle, the recycling of GSSG into GSH, via degradation of the former into amino acids that assembled into GSH, was also documented [30,31]. Supporting this point of view, our results reflected the upregulation of *LAP* and *AP-N* genes encoding leucyl aminopeptidase and aminopeptidase N, which is involved in GSSG degradation [30] as a response to salt stress. In agreement with our results, Amini and Ehsanpour [79] documented the upregulation of the *LAP* gene in tomato leaves exposed to salt stress. The increase in *LAP* transcript abundance was also recorded in response to oxidative stress associated with drought in *Larix olgensis* [80].

The current results reflected increases in *SPS* and *RDS-M* genes, encoding trypanothione synthetase and ribonucleotide reductases, respectively, in response to salt stress. These observations can be explained in light of the involvement of GSH in T(SH)2) synthesis catalyzed with trypanothione synthetase [31] and the role of T(SH)_2_ in H_2_O_2_ detoxification, with simultaneous indirect synthesis of deoxyribonucleotide catalyzed with ribonucleotide reductases [31,32]. In agreement with our results, the *SPS* gene was upregulated in response to salt stress in *Medicago truncatula* [81], Gobindobhog rice [82], and *Citrus aurantium* [83]. Similar to our results, the upregulation of the *RDS-M* gene, accompanied with increase in lipid peroxidation, was recorded in *Larix olgensis* under drought stress [80].

The results recorded for the expression rate of GSH-metabolism-related genes reflected greater transcript abundance for these genes in P plants at the end of the priming period and continued after the stress had vanished. In other words, salt priming was accompanied with irreversible upregulation of the studied GSH-metabolism-related genes. These results confirmed the involvement of GSH metabolism in plant response during salt priming and the subsequent role of priming in improving salt tolerance, which can be understood in light of the previously discussed role of the addressed genes during salt stress.

There is no available literature on the impact of salt priming on the expression of the studied genes. However, Souza et al. [84] documented the upregulation of the *APX* gene in *Physalis angulate* following seed priming with PEG. The stimulatory role of salt priming for genes related to salt tolerance was also recorded by Kubala et al. [85] in *Brassica napus*. The authors observed an increase in the transcript abundance of the gene encoding pyrroline-5-carboxylate synthase A, a key enzyme for proline synthesis.

Our results also reflected a decrease in the expression of the addressed genes in P/Shk plants during salt shock, compared with the Shk plants. The conflict between these results and the defensive role of the corresponding enzymes against salt stress may highlight the exploitation of the priming-induced irreversible upregulation of the studied genes in establishing a stock of these enzymes before the incidence of salt shock in primed plants. Experiencing stress, the primed plant does not need extensive transcription to finalize its partially established defense arsenal. Supporting this point of view, our results reflected a higher APX activity in primed shocked plants, compared with unprimed shocked plants, despite the contrast records for *APX* gene expression. Similar results were recorded for proline-metabolism-related enzymes and their encoding genes in *Brassica napus* [85]. In a similar context, Souza et al. [84] recorded that the beneficial impact of PEG priming during salt stress was not correlated with significant changes in the expression rate of several salt-tolerance-related genes, including the *APX* gene, in *Physalis angulate*.

The decrease in the water potential of the soil solution under salt stress hinders water absorption and decreases the relative water content of the stressed plants. Therefore, to avoid dehydration, the plant accumulates compatible solutes to decrease the water potential of the cell sap, enforcing water absorption from the salt-containing soil solution [16]. The results presented in the current study showed the increase in compatible solutes exemplified by soluble sugars and proline as components of the plant response to salinity stress. Proline acts as a non-toxic osmotic solute, and is able to preferentially maintain the structure of cellular organelles and macromolecules. It is documented that proline accumulation protects plants against osmotic stressors by stabilizing several functional units, such as membranes, complex II electron transport, and proteins including enzymes [86]. In harmony with these results, an increase in proline accumulation was documented under salt stress in faba bean [44], wheat [11], *Pisum sativum* [18], and pearl millet [61].

Our study reflected the accretion of soluble sugars in response to salt treatment at the end of the priming period. Meanwhile, a significant decrease in soluble sugar content was detected at the stress shock period. The early accumulation of soluble sugar content aids the plant in accommodating the osmotic balance, scavenging ROS, and affording energy to endure salt stress [18,87,88]. However, the subsequent decline in soluble sugar accumulation may be attributed to the inhibition of photosynthesis, as well as the metabolic shift that strengthens the biosynthesis of secondary metabolites that are crucial for salinity tolerance [89]. The reduction in soluble sugar accretion in response to salinity was also recorded in *Vicia faba* [89,90].

The current results demonstrated the ability of salt priming to enhance the accumulation of TSS and proline in P plants. The latter remained higher than control, even after stress had been discontinued, while the former returned to the untreated control level. These results were also recorded for sorghum [56,68], melon [69] and wheat [57], confirming the retention of high levels of osmoprotectants as a mechanism and explaining the role of salt priming in the enhancement of the subsequent tolerance to salt stress [67]. The importance of proline for osmotic adjustment in primed plants over TSS was confirmed in the present study by the higher level of proline in P/ShK plants, compared with Shk plants, while TSS were comparable in both plant groups. Similar observations were recorded in wheat [57] and *Mesembryanthemum crystallinum* [58]. Collectively, Figure 9 provides a mechanistic summary of how the priming can help the plants in executing the defense mechanisms to cope with the consequences of salt shock and its injurious effects on growth and yield.

## 4. Materials and Methods

### 4.1. Plant Material and Growth Conditions

Faba bean seeds (*Vicia faba* cv. Sakha 1) were surface-sterilized in sodium hypochlorite (1%) for 2 min, then rinsed three times with distilled water and sown in 30 cm diameter plastic pots filled with a mixture of clay, sand, and peat moss in 1:1:1 (v:v:v). The pots were arranged in the open air at the experimental farm of the Agricultural Genetic Engineering Research Institute (AGERI), Agricultural Research Center (ARC), Giza, Egypt, during the winter season of 2020/2021.

### 4.2. Salt Treatment

The seeds were irrigated with tap water for 31 days; then, three uniform plants per pot were selected for further experimental work in five groups and pots were arranged in randomized complete block design with three replicates per treatment. A preliminary experiment indicated the complete yield inhibition in response to continuous exposure to 200 mM NaCl. The yield-inhibitory concentration (200 mM NaCl) was used to establish salt shock, while a slightly lower concentration (150 mM) was used in priming and positive control (Figure 10) to avoid yield inhibition.

**NC:** irrigated with tap water (negative control).**PC:** irrigated with 150 mM NaCl solution (positive control).**P:** salt primed for 15 days divided into three equal intervals; plants were irrigated with 50 mM NaCl solution in the first interval (5 days), then the salt concentration was increased by 50 mM in each of the next intervals. Thereafter, the salt solution was replaced with tap water till the end of the season.**Shk:** Exposed to 200 mM NaCl for 7 days starting 51 days after sowing (DAS).**P/Shk:** Primed as detailed in P, then shocked as described in Shk.

All groups were fertilized by NPK solution after 15 DAS. The soil was leached, for 5 days, using tap water after 46 DAS to remove accumulated ions. In all treatments, plants were irrigated with 250 mL of their corresponding irrigation solutions per time.

### 4.3. Plant Sampling

Representative samples of leaves were collected from each treatment at 46, 51, and 58 DAS, then powdered in liquid nitrogen and stored at −80 °C for use in physiological analyses in fresh tissues and gene expression analysis. Other representative samples of leaves were collected at 46 and 58 DAS, then dried at 65 °C till weight was constant and saved for analyses in dry tissues. Samples for growth measurements were collected at 58 DAS. At the experiment’s end, pods of each plant were harvested and counted. The seeds were collected from the pods and weighed to calculate the average weight of 1000 seeds and total yield.

### 4.4. Physiological Analysis

#### 4.4.1. Photosynthetic Pigments

Twenty milligrams of fresh leaves were ground in 2 mL of pre-chilled 80% (*v*/*v*) acetone; the clear supernatant was read at 440.5, 644, and 662 nm using a microplate reader (Infinite 200 PRO, Tecan Group Ltd., Männedorf, Switzerland) [91]. Chlorophyll a, chlorophyll b, and carotenoid concentrations were calculated using MacKinney equations, as mentioned in [92].

#### 4.4.2. Determination of Peroxide

The hydrogen peroxide content was determined as described by [93]. Briefly, 150 mg of frozen leaf material was directly homogenized with 1 mLof 0.1% (*w*/*v*) TCA containing 1 M KI and potassium phosphate (KP) buffer (10 mM, pH 5.8) for 10 min. Then, 200 µL of the clear supernatant was incubated for 20 min and the absorbance was recorded. The result was expressed as µmol H_2_O_2_ g^−1^ FW based on an H_2_O_2_ standard curve.

#### 4.4.3. Malondialdehyde (MDA)

MDA levels were estimated according to [94]. Briefly, 20 mg of frozen leaf tissues were homogenized in 500 µL of 0.1% (*w*/*v*) trichloroacetic acid (TCA) at 4 °C. After centrifugation, 50 µL of supernatant was incubated with 150 µL of 0.5 % (*w*/*v*) thiobarbituric acid (TBA) in 20% (*w*/*v*) TCA at 95 °C for 1 hr to generate TBA-MDA. The absorbance was recorded at 532 and 600 nm and the MDA content was measured, after subtraction of non-specific absorbance, as nmol g^−1^ FW using an extinction coefficient of 155 mM^−1^ cm^−1^.

#### 4.4.4. Membrane Stability Index (MSI)

A total of 0.2 g of a freshly collected leaves were incubated in 10 mL of double-distilled water at 40 °C for 30 min, then the electrical conductivity (C_1_) of the solution was recorded using an electrical conductivity meter (Orion 5 Star, Thermo Fisher Scientific, Waltham, MA, USA). In another test tube, the second electrical conductivity (C_2_) was recorded after 10 min incubation at 100 °C. The MSI was calculated using the following formula [52]: MSI (%) = [1 − (C_1_/C_2_)] × 100.

#### 4.4.5. Phenolic Content

Polyphenols were extracted from dried leaf samples in 70% (*v*/*v*) ethanol, according to [95]. Thereafter, 250 µL of the extract was hydrolyzed with 250 µL 2 N HCl in a boiling water bath for 1 h. After neutralization with 0.5 N NaOH, the total phenolic content was estimated at 650 nm following Lowe [96] method and was expressed as mg g^−1^ DW, compared with the gallic acid standard curve.

#### 4.4.6. Antioxidant Enzymes Assay

The activities of guaiacol peroxidase (POD; EC: 1.11.1.7), catalase (CAT; EC: 1.11.1.6), superoxide dismutase (SOD; EC 1.15.1.1), glutathione reductase (GR; EC: 1.6.4.2), and ascorbate peroxidase (APX; EC: 1.11.1.11) were measured according to the method outlined by Grace and Logan [97], with some modifications. One hundred mg of frozen leaves powder was homogenized with 1 mLice-cold potassium phosphate (KP) buffer extraction buffer (50 mM, pH 7.0) containing 0.1 mM EDTA, 1mM ascorbate, 2% (*v*/*v*) glycerol, and 4% (*w*/*v*) polyvinylpolypyrrolidone (PVP). The homogenate was centrifuged at 10,000× *g* for 10 min at 4 °C and supernatant was collected for enzymes assay.

POD activity was determined according to [98] by measuring the rate of tetraguaiacol formation in a 3 mL reaction mixture containing KP buffer (50 mM, PH 7.0), 0.1 mM EDTA, 100 mM guaiacol, 30 mM H_2_O_2_, and 100 µL of crude enzyme extract at 465 nm, based on a molar extinction coefficient (ε) of 26.6 mM^−1^ cm^−1^. The activity was expressed as µmol tetraguaiacol mg^−1^ protein min^−1^.

CAT activity was assayed by monitoring the rate of H_2_O_2_ (ε = 43.6 M^−1^cm^−1^) decomposition at 240 nm in a mixture of 3 mLKP buffer (100 mM, PH 7.0) containing 25 mM of H_2_O_2_ mixed with 100 µL of crude enzyme extract [99]. CAT activity was demonstrated as mmol H_2_O_2_ mg^−1^ protein min^−1^.

SOD activity was estimated as described by [100], depending on its ability to inhibit nitro blue tetrazolium (NBT) photoreduction in a mixture of 10 µL of crude extract mixed with 200 µL of reaction mixture (50 mM KP buffer (pH = 7.8), 0.1 mM EDTA, 13 mM methionine, 60 mM riboflavin, and 75 µM NBT) at 560 nm against a non-illuminated blank. One unit of SOD inhibited 50% of NBT photoreduction.

APX activity was measured according to [101] by monitoring the rate of decrease in oxidized ascorbate with enzyme extract at 290 nm in a mixture of 50 mM KP (pH 7.0) containing 0.5 mM of ascorbic acid, 0.1 mL of H_2_O_2_, and 0.1 mM of EDTA, based on ε of 2.8 mM^−1^ cm^−1^ for ascorbate.

GR activity was assessed by monitoring the rate of decrease in absorbance at 340 nm in a mixture of the crude enzyme extract and KP buffer (50 mM, pH 7.0) containing 3 mM of EDTA, 0.1 mM of NADPH (ε = 6.22 mM^−1^ cm^−1^), and 1 mM of GSSG [102].

The content of total soluble protein was quantified in crude enzyme extract according to Lowry [103], and all enzymes activities were related to each mg of protein.

#### 4.4.7. Total Soluble Sugar (TSS) Content

For the extraction of TSS, dry leaves powder was incubated in 1.5 mL of distilled water at 95 °C for 1 h [104]. The TSS content was determined using the anthrone method, according to [105], based on a standard curve of glucose.

#### 4.4.8. Proline Content

For proline determination, 10 mg of dry leaves material was homogenized in 3% (*w*/*v*) aqueous sulfosalicylic acid, as described in [106] and outlined by [107]. The cooled supernatant was mixed with a 1.25% (*w*/*v*) ninhydrin reagent and quantified at 508 nm based on a proline standard curve, then expressed as mg proline g^−1^ DW.

### 4.5. Gene Expression Analysis

Real-Time PCR Analysis of Glutathione metabolism genes

The faba bean leaves stored in liquid nitrogen were utilized for RNA extraction with a Qiagen RNeasy^®^ Plant mini kit (Qiagen, Hilden, Germany), while first-strand cDNA was synthesized from 0.5 µg of total RNA using a RevetAid First Strand cDNA Synthesis Kit (Thermo Fisher Scientific, USA), following the manufacturer’s instructions.

The genes related to the glutathione metabolism pathways (*L-APX*, *SPS*, *LAP*, *AP-N*, and *RDS-M* genes) were quantitatively amplified with the aid of StepOnePlus (Applied Biosystems®, Thermo Fisher Scientific, USA) Real-Time PCR system using specific primers (Table 1) and a SYBR^TM^ Green PCR master mix (Applied Biosystems®, Thermo Fisher Scientific, USA). The transcription level of the *Vicia faba* Elongation Factor-alpha (*Vf*EFα) gene was used as an endogenous control, to which the transcription level of the addressed genes were normalized using the 2^−DDCt^ method. The amplification protocol consisted of 40 amplification cycles (95 °C for 15 s and 60 °C for 60 s) preceded by 95 °C for 10 min. For each gene, expression estimated in Ctrl was used as a quantification unit.

### 4.6. Statistical Analysis

The results were expressed as the mean value of three replicates with a calculated standard error (±SE). The significance of the difference between mean values was determined at (*p* < 0.05) using a one-way analysis of variance (ANOVA). The mean values were compared through a Duncan multiple comparison test at the 5% level of significance.

## 5. Conclusions

In conclusion, salt stress impairs the growth and yield of broad bean plants, due to the accompanying ionic toxicity, osmotic stress, and accumulation of toxic ROS, injuring cellular functional integrity and causing the oxidation of cell macromolecules, including chlorophyll, proteins, and lipids. Priming is a potent tool for enhancing salt tolerance by maintaining a permanent alert of defense mechanisms that harden the plant against the upcoming stress. The alerted mechanisms enhance the expression of defense-related genes (e.g., GSH metabolism-related genes), proline accumulation, and antioxidant enzymes, which establish a solid defensive front line that counteracts the osmotic and oxidative consequences of salt stress and ameliorates its injurious effect on growth and yield. Thus, the current results open the door for wide application of salt priming to mitigate the injurious effects of salt stress and support food security in the adverse conditions created with climate change.

## Figures and Tables

**Figure 1 plants-11-01610-f001:**
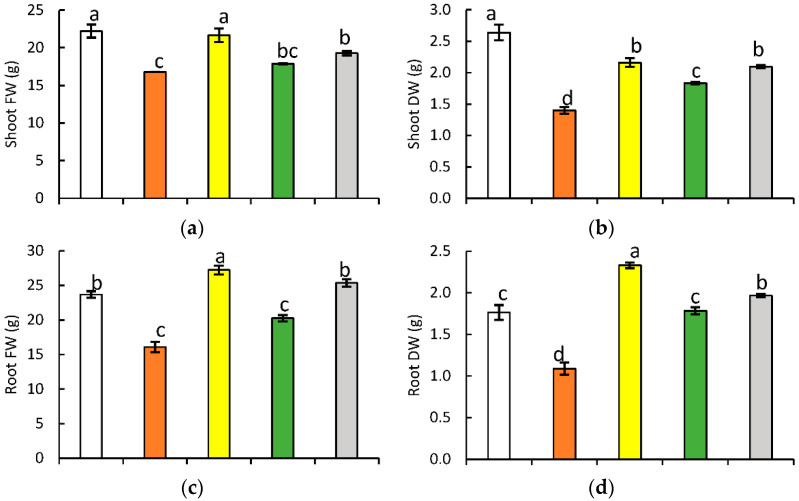
Effect of different salt treatments (NC: negative control, PC: positive control, P: salt priming, Shk: salt shock, and P/Shk: salt priming then salt shock) on shoot fresh weight (**a**), shoot dry weight (**b**), root fresh weight (**c**), root dry weight (**d**), leaf area (**e**), plant height (**f**), pod length (**g**), pods weight/plant (**h**), the weight of 1000 seeds (**i**), and the seed yield/plant (**j**) of faba bean plants. Growth traits were recorded at 58 DAS, while yield was recorded at 100 DAS (days after sowing). The values represented in the figure indicate the mean of three replicates (±SE). Different letters on the error bars indicate significant differences among salt treatments (*p* < 0.05).

**Figure 2 plants-11-01610-f002:**
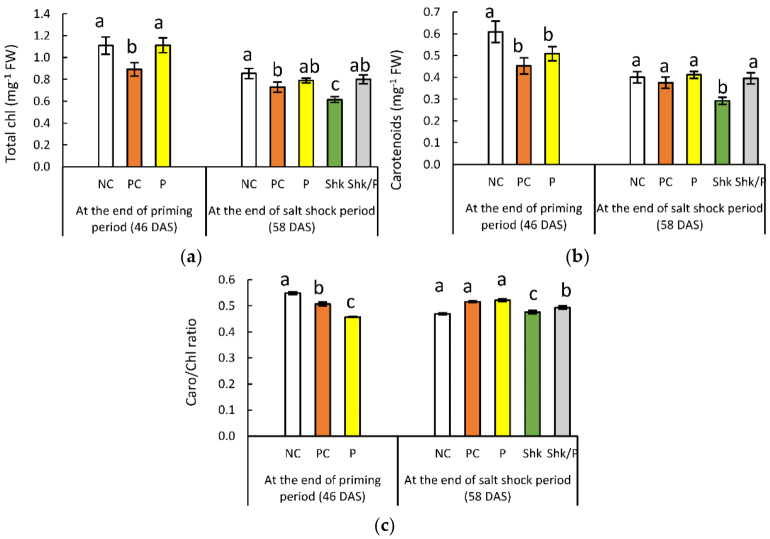
Effect of different salt treatments (NC: negative control, PC: positive control, P: salt priming, Shk: salt shock, and P/Shk: salt priming then salt shock) on (**a**) total chlorophyll, (**b**) carotenoids, and (**c**) the carotenoids/total chlorophyll ratio (C). The values represented in the figure indicate the mean of three replicates (±SE). Different letters on the error bars indicate significant differences among salt treatments (*p* < 0.05).

**Figure 3 plants-11-01610-f003:**
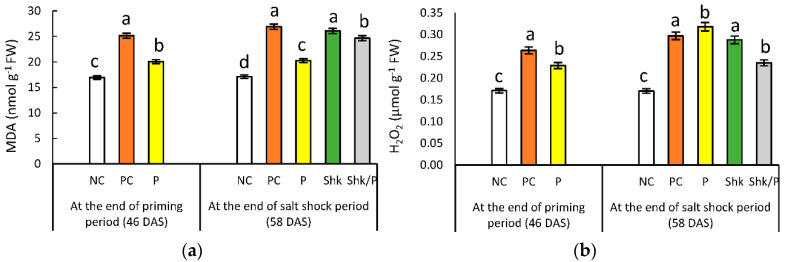
Effect of different salt treatments (NC: negative control, PC: positive control, P: salt priming, Shk: salt shock, and P/Shk: salt priming then salt shock) on (**a**) the MDA level (nmol g^−1^ FW) and (**b**) H_2_O_2_ content (µmol g^−1^ FW). The values represented in the figure indicate the mean of three replicates (±SE). Different letters on the error bars indicate significant differences among salt treatments (*p* < 0.05).

**Figure 4 plants-11-01610-f004:**
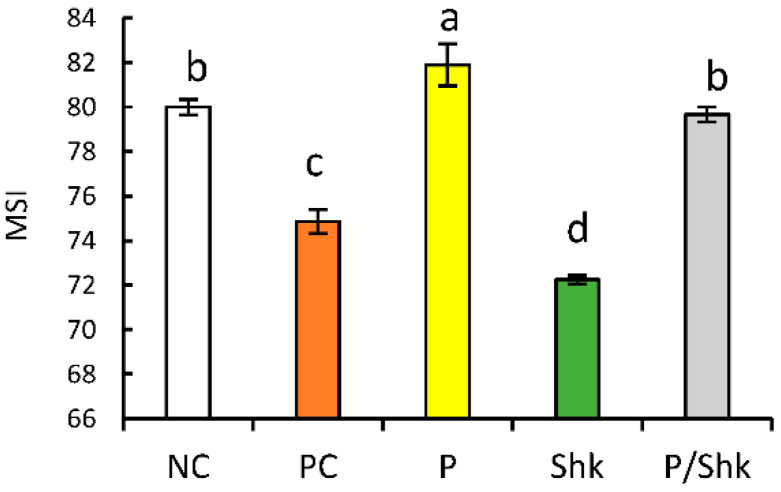
Effect of different salt treatments (NC: negative control, PC: positive control, P: salt priming, Shk: salt shock, and P/Shk: salt priming then salt shock) on the membrane stability index (MSI) at 58 DAS. The values represented in the figure indicate the mean of three replicates (±SE). Different letters on the error bars indicate significant differences among salt treatments (*p* < 0.05).

**Figure 5 plants-11-01610-f005:**
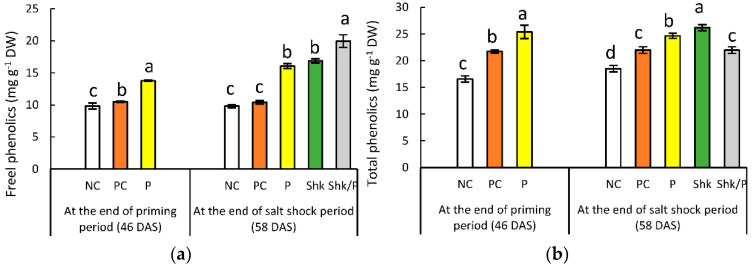
Effect of different salt treatments (NC: negative control, PC: positive control, P: salt priming, Shk: salt shock, and P/Shk: salt priming then salt shock) on (**a**) free phenolics (mg g^−1^ DW), and (**b**) total phenolics (mg g^−1^ DW). The values represented in the figure indicate the mean of three replicates (±SE). Different letters on the error bars indicate significant differences among salt treatments (*p* < 0.05).

**Figure 6 plants-11-01610-f006:**
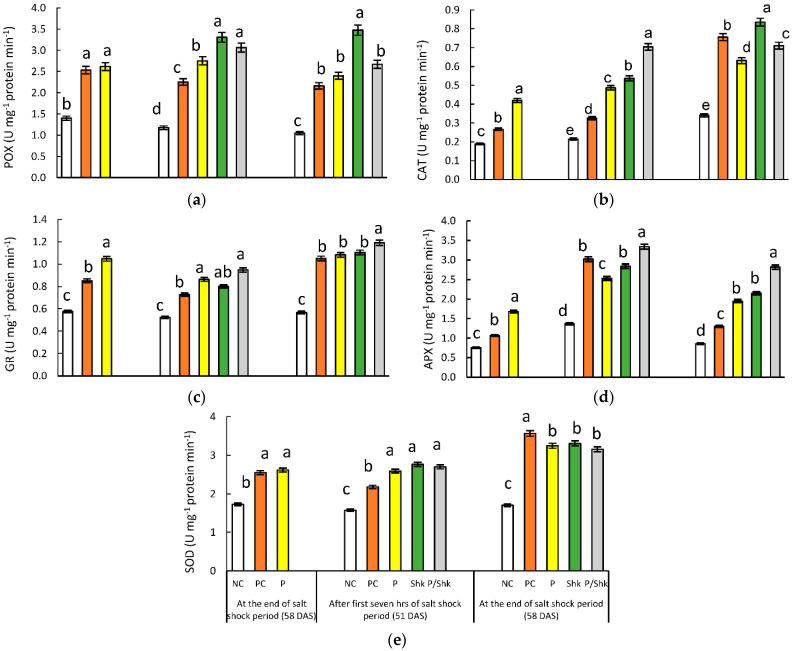
Effect of different salt treatments (NC: negative control, PC: positive control, P: salt priming, Shk: salt shock, and P/Shk: salt priming then salt shock) on (**a**) POX, (**b**) CAT, (**c**) GR, (**d**) APX, and (**e**) SOD activities (U min^−1^ mg^−1^ protein). The values represented in the figure indicate the mean of three replicates (±SE). Different letters on the error bars indicate significant differences among salt treatments (*p* < 0.05).

**Figure 7 plants-11-01610-f007:**
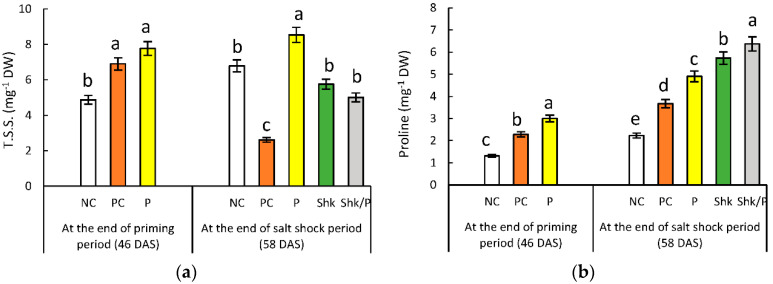
Effect of different salt treatments (NC: negative control, PC: positive control, P: salt priming, Shk: salt shock, and P/Shk: salt priming then salt shock) on (**a**) total soluble sugar (mg g^−1^ DW) and (**b**) proline content (mg g^−1^ DW). The values represented in the figure indicate the mean of three replicates (±SE). Different letters on the error bars indicate significant differences among salt treatments (*p* < 0.05).

**Figure 8 plants-11-01610-f008:**
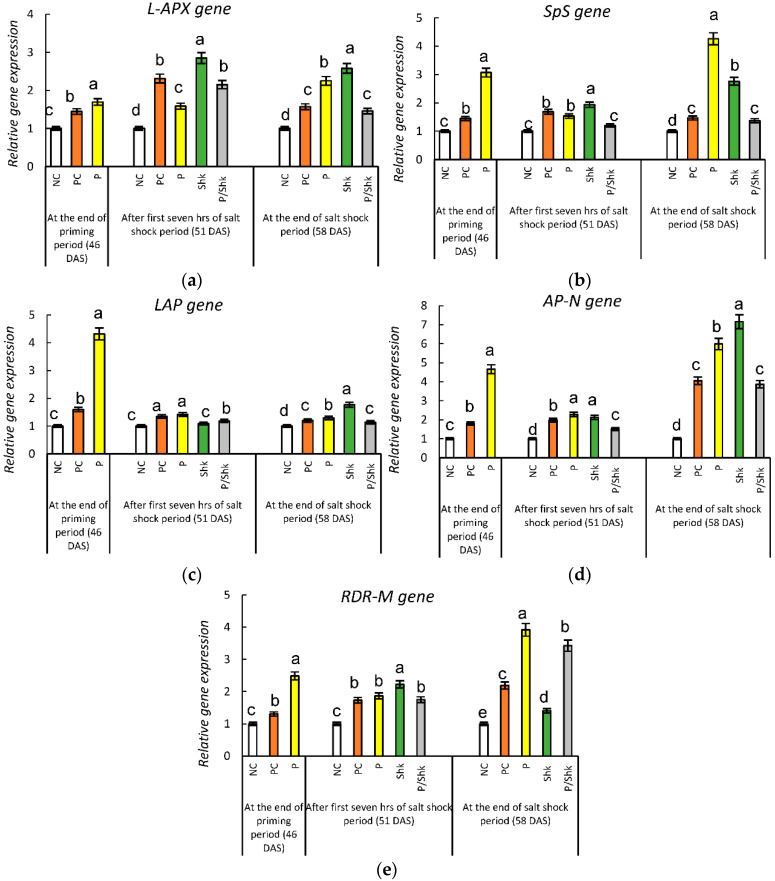
Effect of different salt treatments (NC: negative control, PC: positive control, P: salt priming, Shk: salt shock, and P/Shk: salt priming then salt shock) on qPCR gene expression of glutathione and ascorbate metabolism genes: (**a**) *L-APX*; L-ascorbate peroxidase, (**b**) *LAP*; leucyl aminopeptidase, (**c**) *SPS*; spermidine synthase, (**d**) *AP-N*; aminopeptidase N, and (**e**) *RDR-M*; ribonucleoside-diphosphate reductase subunit M1 gene. The values represented in the figure indicate the mean of three replicates (±SE). Different letters on the error bars indicate significant differences among various salt treatments (*p* < 0.05).

**Figure 9 plants-11-01610-f009:**
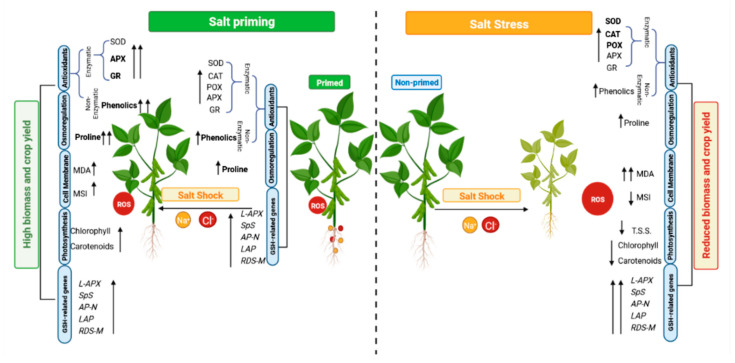
A mechanistic summary showing how priming can alter plant settings to cope with the negative impact of salinity.

**Figure 10 plants-11-01610-f010:**
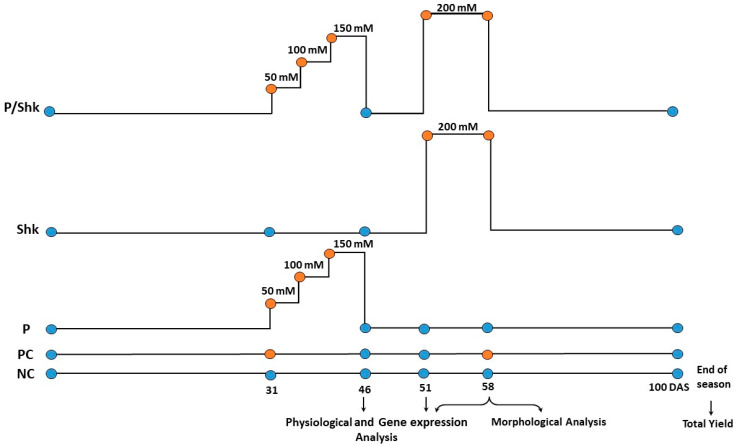
Scheme of the salt priming experimental layout.

**Table 1 plants-11-01610-t001:** Name and sequence of primers used in the qRT-PCR analysis.

Gene	Primer Sequence (5′–3′)		References
	Forward	Reverse	
** *VfEFα* ** **(Housekeeping gene)**	GACAACATGATTGAGAGGTCCACC	GGCTCCTTCTCAATCTCCTTACC	[108]
***L-APX* ** **gene**	ACCGTATTGTGCAGTGCTCA	TTAGCCCTCTCCTGCTGCTA	[109,110]
***SpS* ** **gene**	TTGGTAGGCACATTCGTCCC	AATGTGGCCTGGAGAAGCTC	
***L-AP* ** **gene**	CCTCAAATCGCCGTCCCTAG	TCTCCGGTGACAGTTCTTGC	
***AP-N* ** **gene**	CCGGTCACCTCCTACTGGTA	GGAAGCCTTCTGCCTCACAT	
***RDR-M* ** **gene**	ATCAGCATGCCATGGTTCCA	CCAATGCTGCGTGTGTTCAA	

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
