# Peer review of "Salt Priming as a Smart Approach to Mitigate Salt Stress in Faba Bean (Vicia faba L.)"

_plants, 2022, doi:10.3390/plants11121610_

Round 1

Reviewer 1 Report

            This article describes the significance of salt priming in Vicia faba in lieu of climate change in the future leading to probable increase in salt concentration of soil.

In order to depict salt priming, the authors have laid out the experiments well. They have used non-irrigated tap water as a negative control and positive control as 150mM NaCl. The priming was carried out in between concentrations of NaCl and shocked at 200mM.

            It would be nice to state the reason for using these concentrations!

            The study is comprehensive and various physiological and molecular parameters have been judged to indicate salt priming of bean. This includes measuring growth, chlorophyll content, several enzyme activities indicative of oxidative stress and defense mechanisms. Further, gene expression analysis using qRT-PCR show mainly increased expression of genes for glutathione metabolism, proline accumulation and antioxidative enzymes. Enhanced expression of these genes indicates that the bean plants are ready/primed to combat salt stress.

            The detailed quality of this study are suitable for publication in Plants.

Reviewer 2 Report

Congratulations! to the authors. They have done a good work with experimental planning and execution. The experimental layout was a very good representation. However, need some things addressed to bring out the essence of the paper

Comments

Why was Faba bean selected? Can the authors highlight the need for studying priming salt in these species?

Can authors provide an image of plants after stress treatment, at the time of morphological analysis i.e 58 days.

Can the authors provide a conclusion figure at the overall high and low of the enzymes and genes upregulated and downregulated, and also the authors provide a mechanistic approach as to how the overall mechanism of how the plants are executing the defense mechanisms might be working. Like one can mention/highlight ion homeostasis, or ROS scavenging etc.

Can the authors comment on the ionic imbalance and relative water content? Since salt stress is a combination of osmotic stress and ionic stress.

Line 82: Space problem. Space problem here and there.

Reviewer 3 Report

The topic is relevant and interesting however the study needs improvements. All detailed comments and suggestions can be found in attached file.
